# Association between Body Mass Index and Survival Outcome in Metastatic Cancer Patients Treated by Immunotherapy: Analysis of a French Retrospective Cohort

**DOI:** 10.3390/cancers13092200

**Published:** 2021-05-03

**Authors:** Laetitia Collet, Lidia Delrieu, Amine Bouhamama, Hugo Crochet, Aurélie Swalduz, Alexandre Nerot, Timothée Marchal, Sylvie Chabaud, Pierre Etienne Heudel

**Affiliations:** 1Department of Medical Oncology, Centre Léon Bérard, 69008 Lyon, France; Laetitia.collet@lyon.unicancer.fr (L.C.); aurelie.swalduz@lyon.unicancer.fr (A.S.); 2Department of Prevention Cancer Environment, Léon Bérard Cancer Centre, 69008 Lyon, France; lidia.delrieu@lyon.unicancer.fr; 3Residual Tumor & Response to Treatment Laboratory, RT2Lab, Translational Research Department, INSERM, U932 Immunity and Cancer, Institut Curie, Paris University, 75005 Paris, France; 4Radiology Department, Centre Léon Bérard, 69008 Lyon, France; amine.bouhamama@lyon.unicancer.fr (A.B.); Alexandre@nerot.net (A.N.); 5Data and Artificial Intelligence Team, Centre Léon Bérard, 69008 Lyon, France; hugo.crochet@lyon.unicancer.fr; 6Department of Supportive Care, Institut Curie, 75005 Paris, France; timothee.marchal@curie.fr; 7Department of Clinical Research and Innovation, Léon Bérard Cancer Centre, 69008 Lyon, France; Sylvie.chabaud@lyon.unicancer.fr

**Keywords:** body mass index, immunotherapy, survival, toxicity, advanced cancer, immune-related adverse events

## Abstract

**Simple Summary:**

Immunotherapy has considerably changed the outcomes of cancer patients. However, only a minority of patients respond to immunotherapy and may suffer toxicity. Moreover, strong predictive and prognostic markers are lacking. The number of overweight and obese individuals is steadily increasing in developed countries. This factor is easy to measure and leads to a chronic inflammatory state. We therefore evaluated the relationships between body mass index, survival, and immune-related adverse events in patients treated by immunotherapy for metastatic cancer. Overall survival was better in patients with a BMI ≥ 25 and in those experiencing toxicity. More than 60% of obese patients experienced toxicity. These results should raise the awareness of physicians concerning the importance of body composition in the management of patients on immunotherapy. Body composition, including lean and fat tissue proportions, could be analyzed by CT scans in this selected population.

**Abstract:**

The response to immunotherapy has been little investigated in overweight and obese cancer patients. We evaluated the relationships between BMI, toxicity, and survival in patients treated by immunotherapy for metastatic cancer. We included metastatic cancer patients treated by immunotherapy between January 2017 and June 2020 at the Centre Léon Bérard. In total, 272 patients were included: 64% men and 36% women, with a median age of 61.4 years. BMI ≥ 25 in 34.2% and 50% had non-small cell lung cancer (*n* = 136). Most received monotherapy, with nivolumab in 41.9% and pembrolizumab in 37.9%. Toxicity, mostly dysthyroiditis, occurred in 41%. Median overall survival (OS), estimated by Kaplan–Meier analysis, was significantly longer for patients with a BMI ≥ 25 than for those with a BMI < 25 (24.8 versus 13.7 months HR = 0.63; 95% CI 0.44–0.92, *p* = 0.015), and for patients experiencing toxicity than for those without toxicity (NR versus 7.8 months, HR = 0.22; 95% CI 0.15–0.33, *p* < 0.001). Adjusted OS was associated with toxicity, and the occurrence of toxicity was associated with sex and histological features but not with BMI. Thus, being overweight and experiencing toxicity was associated with longer overall survival in patients treated by immunotherapy. More attention should be paid to body composition in the care of cancer patients.

## 1. Introduction

The development of immune checkpoint inhibitors, such as anti-PD-1, anti-PDL-1, and anti-CTLA4 antibodies, has led to significant changes in the management of patients with certain types of cancer over the last decade, improving overall survival (OS) [1,2]. However, not all patients respond to immunotherapy, and many suffer toxicity [3]. Most immune-related adverse events (irAEs) are reversible, but for patients suffering from more serious side effects (grade ≥ 3), a temporary interruption or permanent cessation of treatment may be considered. This may have a negative impact on tumor response, decreases the chances of controlling neoplastic progression and survival. Moreover, no biological or clinical predictive markers have been validated other than PDL1 expression, which is widely used in clinical practice but has limited predictive value. New markers of immunotherapy efficacy and toxicity are, therefore, urgently required.

The proportion of the population overweight (body mass index, BMI > 25) or obese (BMI > 30) has been steadily increasing in recent decades, particularly in developed countries [4,5]. Recent epidemiological studies have revealed a heavy burden of obesity- and body fat mass-related cancers [6,7]. Obesity, which is characterized by a chronic inflammatory state [8], provides an interface between metabolic and immune cells [9] and modifies the tumor microenvironment. Indeed, preclinical studies have shown that diet-induced obesity can suppress antitumor immunity by decreasing the proportion of naïve CD8^+^ T cells and increasing the size of the myeloid cell population in tumors [10,11]. However, several recent clinical studies have demonstrated a paradoxical positive association of being overweight or obese with a better response and overall survival on immunotherapy, despite a higher frequency of irAEs [12,13,14,15,16]. Nevertheless, these studies were heterogeneous and often limited to melanoma, non-small cell lung cancer (NSCLC), and renal cell carcinoma [17]. Moreover, discrepancies between studies, particularly as concerns of obesity and irAEs [12,13,16], have highlighted the need for further explorations of the association between BMI and outcome on immunotherapy.

In this study, we investigated the relationships between BMI, toxicity, and survival in patients treated by immunotherapy for metastatic cancer.

## 2. Materials and Methods

### 2.1. Study Design and Cohort Population

The cohort was selected with ConSoRe, a new data analysis solution aggregating diverse forms of structured and unstructured data extracted from digital medical files at a number of French cancer centers. ConSoRe uses natural language processing to search aggregated data and perform advanced data mining [18]. This data mining tool was used to find all over the age of 18 years with inoperable locally advanced or metastatic solid tumors who received immunotherapy in metastatic setting between January 2017 and 30 June 2020 at the Centre Leon Berard (Lyon, France). We excluded patients treated with immunotherapies other than pembrolizumab, nivolumab, atezolizumab, durvalumab, avelumab, or ipilimumab in monotherapy or with combinations of ipilimumab with nivolumab or durvalumab with tremelimumab. Patients receiving adjuvant or neoadjuvant immunotherapy and patients receiving immunotherapy in combination with other antitumor therapies (i.e., chemotherapy or targeted therapy) were also excluded from the cohort.

Data collection was approved in September 2020, by the local data protection officer, on behalf of French regulatory authorities (Commission Nationale de l’Informatique et des Libertés, CNIL) in accordance with MR004 methodology. All patients were informed of the possibility of their health data being used for research purposes and expressed no opposition to this possibility.

### 2.2. Data Collection

#### 2.2.1. Patient Characteristics

Clinical data and tumor characteristics were extracted from digital medical files, including hospitalization and consultation notes. The clinical variables extracted included sex, age, weight at diagnosis and before starting immunotherapy, and height. The tumor characteristics extracted included histology, PDL-1 status (considered positive if >1%), microsatellite status (MSS or MSI-H), and the number of previous treatments. Immunotherapy was also dichotomized into two classes for further statistical analysis: anti-CTLA-4 and anti-PD-1/PDL-1 treatments.

#### 2.2.2. Body Mass Index

BMI was calculated as weight in kilograms divided by height (in meters) squared. Patients were classified on the basis of BMI as being (<18.5), normal weight (18.5–24.9), overweight (25–29.9), or obese (≥30), in accordance with World Health Organization (WHO) guidelines [19]. We also dichotomized the baseline BMI category, using a threshold of 25.

#### 2.2.3. Toxicity

Cases of toxicity were identified by examining all hospitalization reports and oncology visit reports recorded in the patient’s medical file. We excluded toxicities unrelated to adverse immune events, such as fatigue. irAEs were classified according to the Common Terminology Criteria for Adverse Events v5.0.

### 2.3. Statistics

The characteristics of the participants are described with means and standard deviations (SDs) for quantitative data and frequencies and percentages for qualitative data. Comparisons between groups (BMI < 25 vs. BMI ≥ 25) were performed with Student’s t-tests and Welsh’s t-test if necessary for continuous variables or Pearson’s χ^2^ tests and Yates-corrected χ^2^ test if necessary for binary or categorical variables.

The primary endpoint of this retrospective study was the assessment of the association between BMI and OS. We calculated OS as the time between the diagnosis of metastatic disease and death from any cause. BMI was initially considered as a dichotomous categorical variable (BMI ≥ 25 and BMI < 25). Kaplan–Meier methods were used to estimate OS, and log-rank tests were used to assess the statistical significance of differences between groups. Time was considered in months.

The second objective of the study was to analyze the impact of toxicity on OS. Kaplan–Meier curves and log-rank tests were used, as described above. Univariate Cox regression models were used to investigate the associations between survival and clinical variables, including the type of primary tumor (non-small cell lung cancer, head, and neck squamous cell carcinoma, digestive cancer, renal cell carcinoma, melanoma, urothelial carcinoma, and others), BMI (<25 vs. ≥25), sex, expression status for CTLA4, PDL1, and MSI, the presence of toxicity, performance status during first-line immunotherapy (0 to 4), and the grade (CTCAE) of the first toxicity observed (1 to 5). Multivariate Cox regression models were generated by a backward approach after testing in univariate models to estimate patient survival in association with BMI adjusted for covariates, including toxicity, performance status during first-line immunotherapy (0 to 4), and MSI. A landmark analysis was then performed to deal with the survivor bias resulting from patients responding to immunotherapy being exposed to immunotherapy for longer, and therefore being more likely to develop irAEs than non-responder patients, in whom immunotherapy was stopped earlier [20]. These analyses considered only toxicities occurring in the first three months of immunotherapy for the classification of groups, with the exclusion of patients who died or were censured before three months of treatment. The proportional hazards hypothesis was tested, and a graphic diagnosis based on the scaled Schoenfeld residuals was performed.

Univariate logistic regression models were then generated to investigate the associations between the occurrence of toxicity and clinical variables. The variables considered included BMI (<25 vs. ≥25), sex, MSI status, number of prior lines of treatment, type of immunotherapy (mono- vs. bitherapy), type of primary tumor (non-small cell lung cancer, head and neck squamous cell carcinoma, digestive cancer, renal cell carcinoma, melanoma, urothelial carcinoma, and others) and CTLA4 expression status. A multivariate analysis was then performed by a backward approach with a subset of these variables after testing in univariate models, including sex and histological features. Odds ratios (ORs) with 95% confidence intervals (CIs) were calculated in both univariate and multivariate analyses.

Statistical analyses were conducted with R software. Values of *p* < 0.05 were considered to indicate significance in all statistical tests.

## 3. Results

### 3.1. Patient Characteristics

The characteristics of the patients are shown in Table 1. In total, 387 patients were treated by immunotherapy for locally advanced or metastatic cancer at the Léon Bérard Cancer Center between January 2017 and 30 June, 2020. We excluded 115 of these patients due to (i) immunotherapy regimens other than those described above (*n* = 13), (ii) immunotherapy in association with chemotherapy or another cancer therapy (*n* = 98), and (iii) other reasons (*n* = 4). The final analysis thus included 272 patients (Figure 1).

Median age at the start of immunotherapy for the patients included was 61.4 years, and 64% of the patients were men (Table 1). Menopausal status was not reported, but only 19 of the female patients were under the age of 50 years. More than 60% of the patients (*n* = 179) had a BMI < 25, whereas 34.2% (*n* = 93) had a BMI ≥ 25. The most frequent type of cancer was NSCLC, in about half the population (*N* = 136), followed by head and neck carcinoma in 13.2% (*N* = 36), and gastrointestinal cancer, renal carcinoma and melanoma, each in 8.5% (*n* = 24, *n* = 24 and *n* = 22, respectively). PDL-1 expression status was positive (>1%) in 75.4% (*n* = 104) of the tumors tested (*n* = 138). Overall, 9% (*n* = 24) of the tumors were MSI-H; 88% of patients (*n* = 239) received monotherapy, with nivolumab in 41.9% (*n* = 114) and pembrolizumab in 37.9% (*n* = 103). Performance status (PS) at the initiation of immunotherapy was ≤1 for 79% (*n* = 215) of patients. In total, 141 toxicity events were reported, in 41.2% of patients (*n* = 112).

The characteristics of the population were compared between the two groups (BMI < 25 and BMI ≥ 25). Age, sex, and performance status (PS) differed significantly between the two groups. The patients with a BMI ≥ 25 were older, more likely to be male (74.2%, *n* = 69), and with a PS of 1 (62.4%, *n* = 58) at treatment initiation. Underweight and normal-weight patients (BMI < 25) mostly had a suitable performance status (PS0 or 1 for 71.5%, *N* = 128) at treatment initiation, and 23.5% (*n* = 42) had a PS of 2, versus 6.4% (*n* = 6) for overweight and obese patients (BMI ≥ 25). The types of primary cancer were evenly similarly distributed between groups, but head and neck carcinoma were not significantly in the most frequent cancer in normal-weight or underweight patients (BMI < 25) (16.2%, *n* = 29) (Table 1), and patients with renal cell carcinoma, melanoma or urothelial carcinoma were not significantly more frequent among overweight or obese patients (BMI ≥ 25) (Table 1).

### 3.2. Characteristics of the Observed Toxicities

In total, 141 toxicity events occurred in 112 patients. Thyroiditis, with either hypo- or hyperthyroidism, was the most common irAE, occurring in 12.5% of patients, followed by rheumatic adverse events (8.8%), diarrhea, and/or colitis (7.4%), and dermatitis (6.6%) (Table 1). Most patients had grade I or II (39.3%) toxicities, with only 12% experiencing a toxicity of grade III or above. Two patients experienced grade 5 toxicities (myocarditis and colitis), causing death after one injection in one patient and after nine months of treatment in the other. Treatment was discontinued due to toxicity in 26% of patients (*n* = 70), and this discontinuation was definitive for 17.3% (*n* = 47). It was temporary in 8.4% (*N* = 23), in whom it was possible to recommence treatment. Toxicity occurred in 58% of patients receiving anti-CTLA4 antibodies (20 patients of 34). However, only 38% of patients treated with PD1 or PDL-1 antibodies experienced toxicity (92 of 238 patients).

### 3.3. Association between BMI and Toxicities

We first considered four classes of BMI (underweight, normal weight, overweight and obese). We found that between 35 and 40% of underweight, normal, and overweight patients experienced immune-related toxicities, whereas such toxicities occurred in 62.5% of obese patients (15 of 24 patients; Appendix A). Moreover, two patients each experienced three toxicity events, and both these patients were overweight.

The types of toxicity observed were globally similar between BMI groups, although more overweight and obese patients experienced pneumonitis (7.5% versus 3.4%) (Table 1). irAEs occurred within a median of 5 months (range, 0.1–28.1), and the time lag to toxicity events was similar between the group of patients with a BMI below 25 and those with a BMI of 25 or above (Figure 2).

### 3.4. Association between BMI and Survival

OS did not differ significantly between the four WHO categories for BMI but tended to be higher in the obesity group (BMI ≥ 30) (*p* = 0.1) (Figure 3A). When BMI was dichotomized, median OS was significantly higher for patients with a BMI ≥ 25, at 24.8 months (95% CI 18.8-NA) versus 13.7 months (95% CI: 8.5 -23.9) for patients with a BMI < 25 (HR= 0.63, 95% CI 0.44–0.92, *p*= 0.015) (Figure 3B).

### 3.5. Association between Toxicity and Survival

Over a median follow-up of 10 months (range: 0.1–48.5), median OS was 21.6 months (95% CI: 13.8–24.8) for the whole population. Median OS was not reached (95% CI 34.5-NA) for patients experiencing toxicities but was 7.8 months (95% CI: 5.4–10.9) for patients without toxicity (HR = 0.22; 95% CI 0.15–0.33, *p* < 0.001) (Figure 4A). The three-month landmark analysis, used for sensitivity analysis, included 221 patients, 28% (*N* = 62) of whom had irAEs, the remaining 72% (*N* = 159) experiencing no toxicity. Median OS was not reached (95% CI: 23–NA) for patients with toxicities, but was 24.4 months (95% CI 19.7–35.8) for patients without toxicity (HR = 0.55; 95% CI 0.32–0.94, *p* = 0.028) (Figure 4B).

### 3.6. Determinants of Survival

In multivariate analysis, the occurrence of toxicity (HR = 0.25 95% CI 0.16–0.38) and MSI-high status (HR = 0.26, 95% CI 0.11–0.63) were associated with prolonged survival. BMI (≥ 25 versus < 25) was no longer associated with OS in multivariate analysis (HR = 0.75 95% CI 0.51–1.10 *p* = 0.14) (Figure 5).

### 3.7. Occurrence of Toxicity

In multivariate analysis, toxicity was found to occur less frequently in men (men versus women OR = 0.53 95% CI 0.31–0.90) and more frequently in patients with melanoma (melanoma versus NSCLC OR = 3.42 95% CI 1.33–9.56) (Figure 6). These results were confirmed by the three-month landmark analysis of sensitivity.

## 4. Discussion

In this retrospective study, overweight and obese patients were found to have longer overall survival than normal-weight or underweight patients.

The median OS of 24.8 months (95% CI 18.8-NA) for this cohort is consistent with the findings of other studies. Indeed, in a recent study of 976 patients, 65% of whom treated for NSCLC, a median OS of 26.6 months (95% CI: 21.4–36.8) was reported, and median OS was higher in patients with a BMI ≥ 25 (HR= 0.33 95% CI: 0.28–0.41) [13]. Most of the other studies demonstrating a relationship between BMI and outcome on immunotherapy [17,20,21,22,23] were also performed on patients with NSCLC [16,24,25], renal carcinoma, or melanoma [12,26,27]. Moreover, two studies, including patients treated by immunotherapy or chemotherapy, demonstrated a prognostic effect of BMI exclusively in patients on immunotherapy. McQuade et al. pooled six independent cohorts of patients with metastatic melanoma treated by chemotherapy, immunotherapy, or targeted therapy; they found a survival benefit in obese patients only in the immunotherapy and targeted therapy cohorts [12]. Cortellini et al. assessed survival outcomes in patients with metastatic NSCLC treated with first-line pembrolizumab or platinum-based chemotherapy; they found that progression-free survival and OS were longer in obese patients exclusively in the immunotherapy subgroup [28]. These results highlight a specific role of obesity in the efficacy of immunotherapy, suggesting that further studies should be performed on this population.

Adipose tissue is associated with a chronic inflammatory state [29] and secretes cytokines, such as adiponectin or leptin, with a direct impact on the tumor microenvironment [30,31]. Indeed, it has an impact on immune cells, causing a decrease in naïve CD8^+^ T-cell levels an increase in the size of the protumoral myeloid cell population in the tumor [10]. In addition, the leptin pathway, which is characteristically overexpressed in obesity [32], may exhaust T cells through PD-1 immune checkpoint overexpression [33]. However, leptin is also known to act on immune system priming, promoting the antitumor immune response through Th1 and Th17 pathway activation together with T-cell differentiation and proliferation [34]. Thus, although the mechanisms involved remain poorly understood, obesity may enhance the immune system but decrease antitumor immune responses mediated, in part, by PD-1 expression, both of which could increase the efficacy of immune checkpoint blockade efficacy [33]. However, further studies are required to improve our understanding of the link between obesity and tumor immune response for the various pathways.

In parallel, a high BMI may also increase the rate of occurrence of irAEs. In our study, 41% of patients experienced irAEs, 75.8% were grade I or II adverse events, consistent with published findings [3,35,36]. Toxicities occurred in 38% of patients with a BMI < 25, versus 46.2% for those with a BMI ≥ 25. Overall, 62.5% of obese patients experienced toxicities. However, multivariate analysis failed to demonstrate an association between BMI and the occurrence of toxicity, probably due at least partly to the small sample size. Obesity has also been associated with higher rates of autoimmune and inflammatory diseases in the general population [37], but studies on patients treated with immunotherapies have yielded discrepant results. Some showed that BMI had no predictive value for the incidence of irAEs [12,16,22,38], whereas others reported a significant increase in the risk of irAEs with increasing BMI [15,39,40,41,42,43,44]. The exact mechanism of this association is not completely understood. Disruption of the immune checkpoint by immunotherapy, leading to immune cell activation and the production of pro-inflammatory cytokines, results in off-target inflammation and autoimmunity [45]. We assume that the pre-existing chronic inflammation in obesity may enhance the Th1/Th17 response and the imbalance in favor of CD8^+^ T cells over T-reg cells in the bloodstream, potentially increasing the risk of autoimmune disease in the general population and of irAEs in patients treated by immunotherapy [46], with the abolition of the inhibition of the negative immune checkpoint increased in obese patients, as previously reported.

The occurrence of irAEs was also associated with a higher OS in the entire population, as confirmed by the sensitivity analysis, which addressed the issue of the bias due to longer periods of exposure to treatment, leading to a higher risk of irAEs in patients responding to immunotherapy. Moreover, the occurrence of irAEs was also found to be associated with overall survival in multivariate analysis. Our results are consistent with those of a large number of studies demonstrating relationships between irAE occurrence and response to treatment or survival outcome [47]. Furthermore, the early onset of irAEs is known to be predictive of response and of better survival on immunotherapy [48,49].

In multivariate analysis, the occurrence of toxicities was associated with OS, but BMI was at the limit of significance. These results may support our hypothesis that obesity improves outcome on immunotherapy through activation of the immune system, as highlighted by the occurrence of toxicities, such that irAEs are more strongly correlated with survival than BMI itself. Furthermore, performance status at the initiation of immunotherapy was ⩽1 for 79% of patients, and 85.7% received immunotherapy as a first-, or second-line treatment, implying that most of our patients were in suitable general condition. These characteristics, together with the low median age of the cohort, are associated with better outcomes and a lower likelihood of undernutrition [50,51] and may have induced a bias concealing the association between higher BMI and survival in multivariate analysis. Moreover, patients with a BMI ≥ 25 also more frequently had a PS of 0 and were less likely to have head and neck carcinoma, and this may also have interfered with the analysis, leading to an overestimation of survival.

BMI determination is recommended for the evaluation of obesity-related health risks [52] but is probably not the most effective way to estimate the amount of fat and its distribution or the level of adipose inflammation [53]. Indeed, visceral adipose tissue is more active than subcutaneous adipose tissue and may be associated with metabolic issues, such as insulin resistance and a chronic inflammatory state [54]. Moreover, metabolic syndrome is known to increase cancer risk and mortality [55] through hyperglycemia, hyperinsulinemia, insulin-like growth factor 1 (IGF-1), or an inflammatory state, and to promote cell growth and division [56]. Subcutaneous adipose tissues are often associated with longer survival in cancer [57], whereas higher visceral adipose tissue levels are associated with a poorer prognosis [58,59,60]. However, little is currently known about the situation in patients on immunotherapy. A recent in vivo study reported a decrease in the sensitivity of tumor cells to ipilimumab and an increase in cardiotoxicity in the presence of high glucose concentrations [61]. Martini et al. also highlighted an association of higher BMI and subcutaneous fat index with prolonged survival on immunotherapy, whereas no such association was observed for high visceral adipose tissue levels [21]. Methods such as waist circumference determination [62] or computed tomography assessments [63] provide results more closely correlated with body composition. Biomarkers of metabolic syndrome are also being investigated [61] and may also be useful for improving the characterization of body composition and outcome on anticancer treatments, including immunotherapy. Furthermore, the percent lean tissue is often low in obese patients [64], potentially affecting pharmacokinetics and exposure to treatment [38,65]. An association between sarcopenic obesity, defined as obesity with severe muscle depletion and survival outcome, has been clearly described for chemotherapy [66,67,68] but has rarely been reported for immunotherapy [44]. However, sarcopenia alone has already been shown to be associated with a poor prognosis in patients on immunotherapy [25,43,44,69,70]. Not only do sarcopenic patients have a poorer survival outcome, they also more frequently suffer irAEs on immunotherapy [15]. Thus, body composition, including factors such as sarcopenia, fat mass and BMI, should be taken into account more carefully in cancer patients, particularly those treated by immunotherapy, to increase the benefits and minimize the risk of treatment and to improve monitoring.

## 5. Conclusions

In conclusion, our findings demonstrate an association of higher BMI with the occurrence of irAEs. Longer overall survival in patients with BMI > 25 treated by immunotherapy was also demonstrated with a 37% reduction in relative risk of death (HR = 0.63). However, in multivariate analysis, BMI was no longer associated with OS. OS was also longer in patients experiencing toxicities. One of the strengths of this study is that it considered the relationships between BMI and both overall survival and irAEs. The diverse range of primary tumors included may have induced a bias, which we tried to deal with by including this variable in uni- and multivariate models. However, the heterogeneity of the population may have enhanced the prognostic and predictive value of BMI across all types of cancer on immunotherapy, a finding of interest given the expansion of immunotherapy to many different types of cancer.

The results should be interpreted with caution, given the retrospective single-center nature of this study, and its small sample size, particularly for subgroup analysis. Studies with a larger sample as well as larger studies in specific tumor types and including patients undergoing the same immunotherapy regimen are now required to explore and confirm these findings.

It may also be relevant to consider and monitor BMI and other body composition parameters in immunotherapy studies. In this way, it may also be possible to identify prognostic and predictive factors and increase the use of multimodal approaches, including nutritional care and physical activities, for example, in cancer management [71].

## Figures and Tables

**Figure 1 cancers-13-02200-f001:**
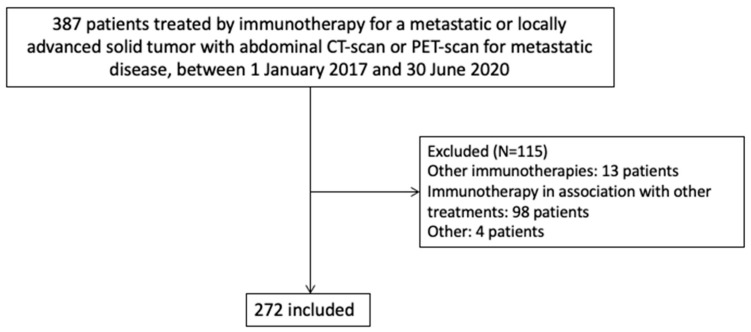
Flow chart of the study population (*n* = 272).

**Figure 2 cancers-13-02200-f002:**
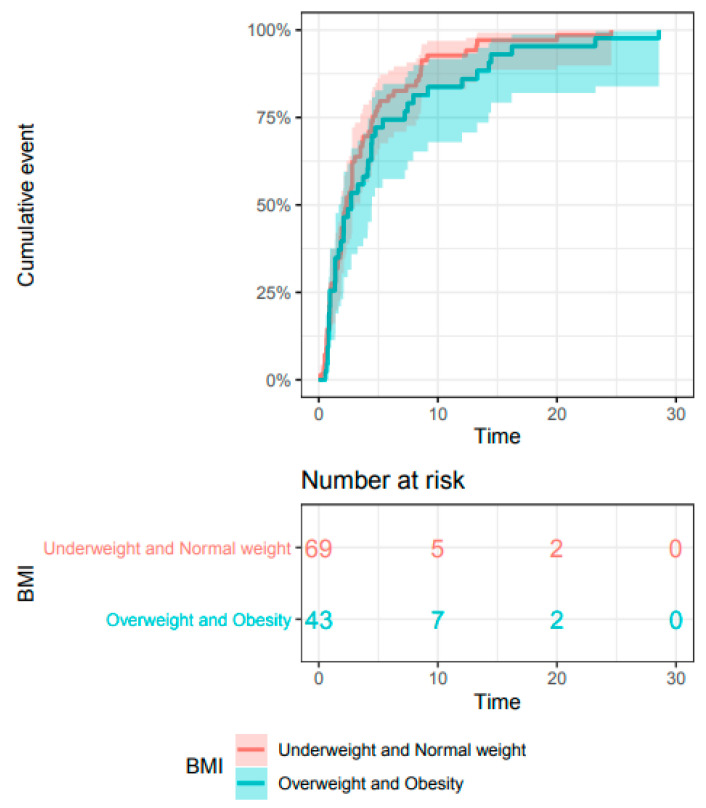
Kaplan–Meier estimate of time to toxicity in the whole population, by BMI category (*n* = 272).

**Figure 3 cancers-13-02200-f003:**
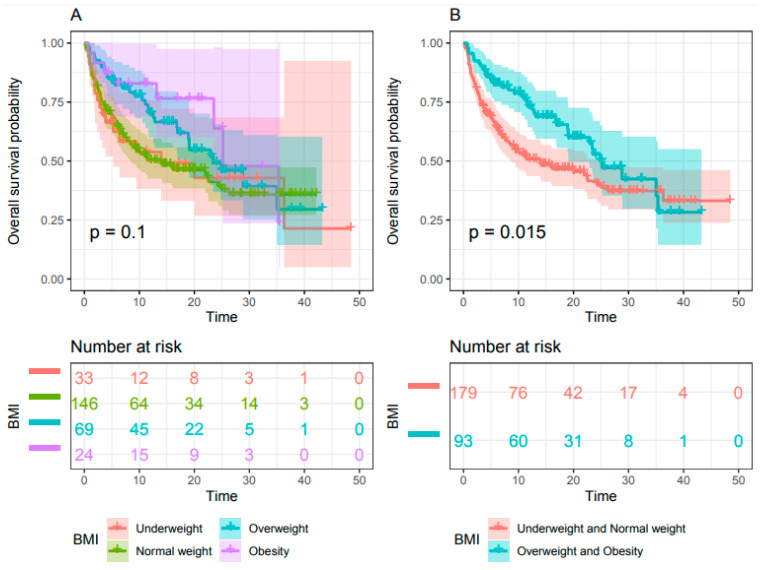
Kaplan–Meier estimate of overall survival. (**A**) Patient BMI classified into the four groups of the WHO classification and (**B**) patient BMI classified into two groups (BMI ≥ 25 versus BMI < 25) (*n* = 272).

**Figure 4 cancers-13-02200-f004:**
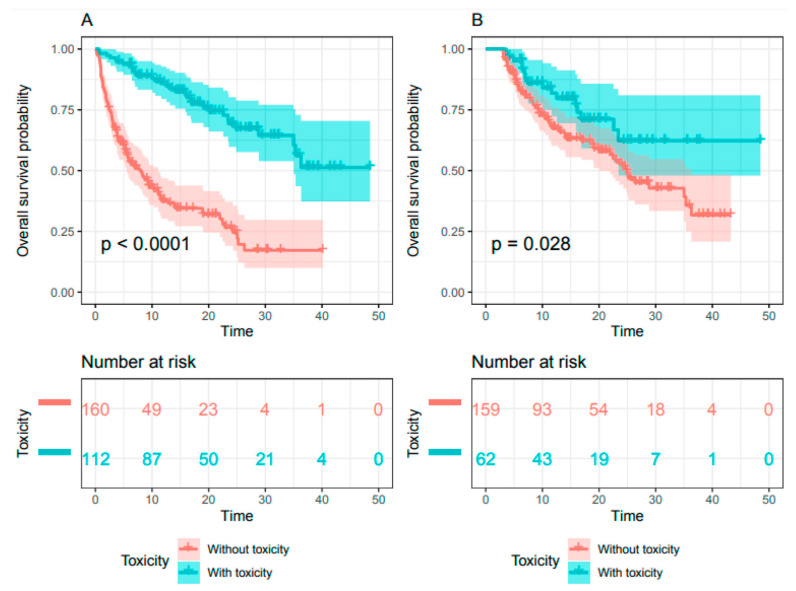
Kaplan–Meier estimate of overall survival for (**A**) the whole population and (**B**) the three-month toxicity landmark (*n* = 272)**.**

**Figure 5 cancers-13-02200-f005:**
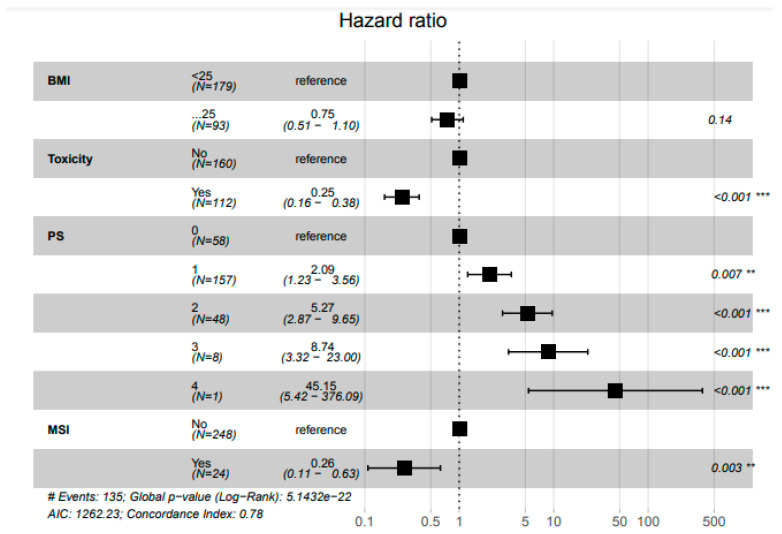
Pooled analysis for overall survival (*n* = 272). ** Statistically significant with *p* < 0.01; *** Statistically significant with *p* < 0.001.

**Figure 6 cancers-13-02200-f006:**
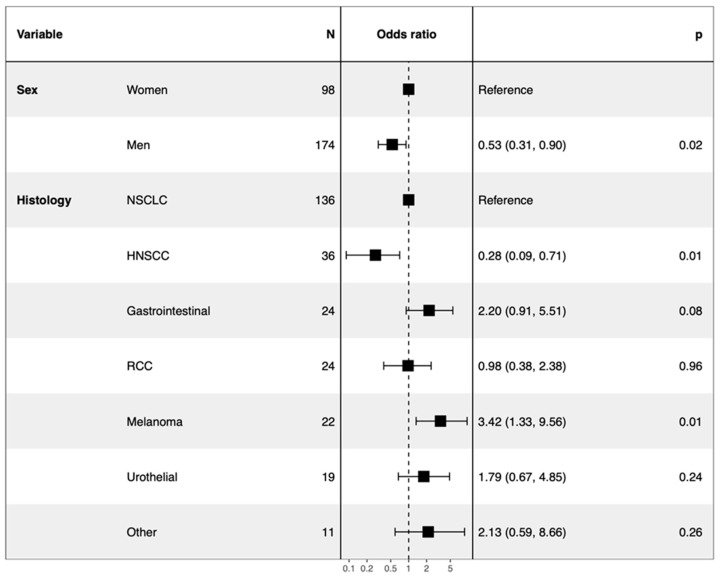
Forest plot of multivariate logistic regression analysis highlighting the independent factors associated with toxicity (*n* = 272).

**Table 1 cancers-13-02200-t001:** Clinical and demographic characteristics: comparison of underweight and normal-weight patients (BMI > 25) with overweight and obese patients (BMI ≥ 25) (*n* = 272).

Clinical Characteristics	Underweight or Normal Weight (BMI < 25) (*n* = 179)	Overweight or Obese (BMI ≥ 25) (*n* = 93)	Overall Population (*n* = 272)	*p*-Value
Age, mean (SD)	59.4 (12.2)	65.3 (9.89)	61.4 (11.8)	<0.001
Sex, *n* (%)				0.02
Men	105 (58.7)	69 (74.2)	174 (64)	
Women	74 (41.3)	24 (25.8)	98 (36)	
Primary tumor, *n* (%)				0.16
Non-small cell lung cancer	93 (52.0)	43 (46.2)	136 (50.0)	
Head and neck squamous cell carcinoma	29 (16.2)	7 (7.5)	36 (13.2)	
Digestive tumor	16 (8.9)	8 (8.6)	24 (8.8)	
Renal cell carcinoma	14 (7.8)	10 (10.8)	24 (8.8)	
Melanoma	12 (6.7)	10 (10.8)	22 (8.1)	
Urothelial carcinoma	9 (5.0)	10 (10.8)	19 (7.0)	
Other	6 (3.4)	5 (5.3)	11 (4.0)	
MSI status, *n* (%)				0.56
MSI-H	14 (7.8)	10 (10.8)	24 (8.8)	
No information or negative	165 (92.2)	83 (89.2)	248 (91.2)	
PDL1 (*N* = 138), *n* (%)				0.48
>1%	73 (40.8)	31 (33.3)	104 (38.2)	
≤1%	22 (12.2)	12 (13)	34 (12.5)	
Treatment, *n* (%)				0.25
Nivolumab	80 (44.7)	34 (36.6)	114 (41.9)	
Pembrolizumab	64 (35.8)	39 (41.9)	103 (37.9)	
Nivolumab + ipilimumab	10 (5.6)	9 (9.7)	19 (7)	
Durvalumab + Tremelimumab	9 (5)	5 (5.4)	14 (5.1)	
Atezolizumab	11 (6.1)	2 (2.2)	13 (4.8)	
Durvalumab	5 (2.8)	2 (2.2)	7 (2.6)	
Ipilimumab	0	1 (1)	1 (0.4)	
Avelumab	0	1 (1)	1 (0.4)	
Performance status, *n* (%)				<0.01
0	29 (16.2)	29 (31.2)	58 (21.3)	
1	99 (55.3)	58 (62.4)	157 (57.7)	
2	42 (23.5)	6 (6.4)	48 (17.6)	
3	8 (4.5)	0	8 (2.9)	
4	1 (1)	0	1 (1)	
Place of immunotherapy in the course of treatment, *n* (%)				0.59
First line	50 (27.9)	31 (33.3)	81 (29.8)	
Second line	99 (55.3)	53 (57.0)	152 (55.9)	
Third line	19 (10.6)	7 (7.5)	26 (9.6)	
>Third line	11 (6.1)	2 (2.2)	13 (4.7)	
Number of toxicities in patients, *n* (%)				0.17
No toxicity	56 (61.5)	50 (53.8)	160 (58.8)	
One toxicity	54 (30.1)	31 (33.3)	85 (31.2)	
Two toxicities	15 (8.4)	10 (10.7)	25 (9.2)	
Three toxicities	0	2 (2.2)	2 (1)	
Type of toxicity, *n* (%)				0.47
Thyroiditis	22 (12.3)	12 (12.9)	34 (12.5)	
Rheumatologic toxicity	14 (7.8)	10 (10.7)	24 (8.8)	
Diarrhea and/or colitis	14 (7.8)	6 (6.4)	20 (7.4)	
Cutaneous toxicity	12 (6.7)	6 (6.4)	18 (6.6)	
Hepatitis	7 (4)	6 (6.4)	13 (4.8)	
Pneumonitis	6 (3.4)	7 (7.5)	13 (4.8)	
Other	9 (5)	10 (10.7)	19 (7)	
Toxicity grade, *n* (%)				0.46
Grade I	31 (17.3)	21 (22.6)	52 (19.1)	
Grade II	32 (17.9)	23 (24.7)	55 (20.2)	
Grade III	15 (8.3)	12 (12.9)	27 (9.9)	
Grade IV	5 (2.8)	0	5 (1.8)	
Grade V	1 (1)	1 (1)	2 (1)	
Immunotherapy discontinued due to toxicity, *n* (%)				1
Temporarily	13 (7.2)	10 (1)	23 (8.4)	
Definitively	27 (15)	20 (21.5)	47 (17.3)	

## Data Availability

The data presented in this study are available on request from the corresponding author.

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
