# Peer review of "Association between Body Mass Index and Survival Outcome in Metastatic Cancer Patients Treated by Immunotherapy: Analysis of a French Retrospective Cohort"

_cancers, 2021, doi:10.3390/cancers13092200_

Round 1
Reviewer 1 Report
The manuscript titled " Association between body mass index and survival outcome in metastatic cancer patients treated with immunotherapy: analysis of a French retrospective cohort" is an interesting research based on the correlation between BMI and responsiveness to ICIs. The manuscript is well conducted, results are clear and well described. Statistical analysis are properly performed. However, some considerations are needed: several authors declaire that BMI is not a proper parameter for determination of risk of mortality or survival; waist circumferance, body fat and visceral fat are a more proper parameters potentially correlated to CDV diseases, risk of mortality and responsiveness to anticancer therapies in patients with cancer. Authors should describe in discussion the putative role of visceral fat and body fat in ICIs responsiveness, considering that some preliminary data indicates that hyperglycemia ( associated to metabolic syndrome) could reduce ICIs anticancer therapies ( you can cite doi: 10.3390/ijms21207802 ). Metabolic syndrome could interfere with anitcancer properties of several drugs, including ICIs ( see doi: 10.18632/oncotarget.16725 ).
The manuscript will be acceptable after minor revision
Reviewer 2 Report
Collet et al. present a retrospective study where they evaluated the relationship between BMI, toxicity and survival in patient treated with immunotherapy for metastatic cancers. This is a single center study, where metastatic cancer patients treated with immunotherapy between January 2017 and June 2020 at Centre Léon Bérard where included.
The cohort of this study is very heterogeneous, but 50% of the patients included have non small cell lung cancer. In the main analyses, patients with different types of cancer were pooled together and as far as I can see, no additional adjustments were made. The groups of patients with tumor types do not seem to be very balanced, and it is very well known that distinct tumor types can have very different prognosis. Overall, I do not think it is appropriate to pool patient cohorts that are so different unless the objective is to merely describe immune-related adverse events and/or their management. It should be also highlighted that nor the association of BMI or irAE with survival are novel concepts, but already shown in multiple studies and summarized in reviews (e.g. obesity in this same journal https://www.ncbi.nlm.nih.gov/pmc/articles/PMC7281442/ ; other extensive studies in NSCLC here as an example https://pubmed.ncbi.nlm.nih.gov/31876896/ and https://pubmed.ncbi.nlm.nih.gov/33119034/). The potential novelties (if any) of this manuscript should be otherwise highlighted
Reviewer 3 Report
The authors present a retrospective study of adult patients with inoperable locally advanced or metastatic cancer treated with immunotherapy, investigating relationship of BMI, toxicity and survival. Median overall survival was found to be significantly longer for overweight and obese patients. However, in multivariate analysis, BMI lost its significance.
The study is of interest in view of growing role and importance of immunotherapy in treatment of solid as well as hematological tumors. The manuscript provides additional knowledge to medical oncologists on the role of body composition specifically for the patients receiving immunotherapy.
There are several places that need clarification:
Line 92. The description of patient cohort is not clear. “Patients receiving immunotherapy as adjuvant /neoadjuvant treatment or in combination with chemotherapy (…) were excluded”. Should it be immunotherapy in combination (…). Please make it more clear which patients were excluded.
Lines 116-117. “For all toxicities we reported (…), type and grade, treatment (…), date of discontinuation (…)”. These characteristics are not included into analysis, subsequently not necessary to include into description.
Line 120. OS is described as the (…) time between diagnosis of metastatic tumor and death of any cause. Did all these patients receive immunotherapy only for their treatment after metastatic disease was established?
Lines 133-135. “a univariate Cox model (…) to calculate hazard ratio for several variable of interest on OS. A clinical model (…) seven risk factors as inputs. Which were the factors that were included into the Cox model, please describe.
Lines 137-139. …multivariate logistic regression model (…). Please describe the variables included into multivariate analysis.
Lines 153-163. Repeats the data that are provided in the Table 1. Would be more important to comment whether there was a difference of characteristics between the BMI groups <25 vs. ≥25.
Lines 175-176. Twenty six percent (…) had to stop treatment (…) and discontinuation was definitive (…). Please clarify the difference between ‘stop’ the treatment and ‘discontinuation’.
Lines 181-182. “Head and neck carcinoma was most common in normal or underweight patients (…). Table 1 shows NSCLC to consist half of the cases (52%) in this group? Please comment.
Lines 185-186. “When we considered BMI in four class, between 35 and 40% of underweight (…), whereas toxicities accounted for 62% of obese patients (15 out of 24), data not showed”. The meaning is not clearly described, also, data not showed. Should not be included into the text.
Table 1. The data are nicely presented however, comparison of characteristics between the groups of BMI <25 vs ≥25 would be of interest.
Not least, revision of the style and English language is necessary. Quite many inaccuracies that make the manuscript difficult to read.
Line 38. “Mostly received immunotherapy (…)”. Most of the patients? Please, correct.
Line 39-40. (…)toxicities, mostly dysthyroidis”. disthyroiditis?
Line 39. “41% experienced (…)” – should not be in numbers in the beginning of sentence.
Line 158. “Among the tumor tested for (…)” – among the tumors.
Line 159. “Nine percent (n=24) tumors (…)” – nine percent of tumors
And many more
Throughout the manuscript, overweight and obese patients are defined in different ways, i.e. BMI < or ≥25, above/below 25 – it is difficult to follow, suggestion to harmonize.
Reviewer 4 Report
sex and menopausal status should be considered among variables since they may affects incidence, prognosis and mortality of cancer in obese patients.
Reviewer 5 Report
This is a retrospective study that shows higher rates of irAEs in obese cancer patient in response to immunotherapy with ICB antibodies. The irAEs observation can be expected; however, it is quite striking that obese patients have longer overall survival (especially those with irAES). This a very important observation that will probably require confirmation in larger/independent cohort of patients. But due to its relevance in the field I recommend publication as it is.
Round 2
Reviewer 2 Report
None
Reviewer 3 Report
Dear Authors,
thank you for reviewing the manuscript.
English language and style is improved, therefore easier to read and understand.
Study design and methods of statistical analysis are better described.
Retropsective origin and very heterogeneous population is a weakness of the study. BMI looses its significance for survival, this needs to be mentioned in the Conclusions section.
OS is described as: "the time between the diagnosis of metastatic 126 disease and death from any cause" (lines 126-127). Did all the patients received immunotherapy only during that period? If not, survival analysis should be adjusted.
